# Stochastic Deep Gaussian Processes over Graphs

**Naiqi Li**[1,*]   **Wenjie Li**[2,*]   **Jifeng Sun**[1]   **Yinghua Gao**[2]   **Yong Jiang**[1,3]   **Shu-Tao Xia**[2,3,†]

[1]Tsinghua-Berkeley Shenzhen Institute, Tsinghua University
[2]Shenzhen International Graduate School, Tsinghua University
[3]PCL Research Center of Networks and Communications, Peng Cheng Laboratory

## Abstract

In this paper we propose Stochastic **D**eep **G**aussian **P**rocesses over **G**raphs (DGPG), which are deep Gaussian models that learn the mappings between input and output signals in graph domains. The approximate posterior distributions of the latent variables are derived with variational inference, and the evidence lower bound is evaluated and optimized by the proposed recursive sampling scheme. The Bayesian non-parametric natural of our model allows it to resist overfitting, while the expressive deep structure grants it the potential to learn complex relations. Extensive experiments demonstrate that our method achieves superior performances in both small size ($< 50$) and large size ($> 35,000$) datasets. We show that DGPG outperforms another Gaussian-based approach, and is competitive to a state-of-the-art method in the challenging task of traffic flow prediction. Our model is also capable of capturing uncertainties in a mathematical principled way and automatically discovering which vertices and features are relevant to the prediction.

## 1   Introduction

Gaussian processes (GPs) [1] are a favourable choice in the machine learning arsenal, due to their distinguishing advantages in modeling uncertainties, the ability of introducing expert knowledge through the flexible kernel design, and the data efficient property which accounts for their success in small and medium datasets. GPs have been successfully applied in a variety of tasks, including computer vision [2], Bayesian optimization [3], active learning [4], multi-task learning [5] and reinforcement learning [6]. However standard GPs scale poorly as $O(N^3)$, making it a challenge to apply them to large-scale datasets. Their expressiveness is also limited by the specific choice of kernel functions, which are difficult to be manually decided for complex problems.

Recently a series of works about deep GPs were presented [7–9], which overcome the aforementioned disadvantages. All these methods can find their roots in the seminal paper [10], which proposes to use a set of inducing points whose size is manageable to summarize all the information in the original large dataset - the inducing points and their latent function values can be inferred by variational inference. This sparse approximation technique enables deep GPs to handle extremely large datasets. For instance [11] reports that their method achieves superior performance on a dataset with over one billion data points. The deep hierarchical structure has greatly improved the expressiveness of the models, resulting in state-of-the-art performances over various challenging tasks.

The last decade also witnesses the rapid development of machine learning algorithms over graph-structured datasets. A great amount of impacting research on graph neural networks (GNNs) has been presented, including [12–17], to name only a few of which. Hitherto most of the emerging literature on graph data analysis are based on neural networks. Though demonstrating satisfactory

---

[*]Equal contributions.
[†]Corresponding author: Shu-Tao Xia (xiast@sz.tsinghua.edu.cn).

performances over various graph learning tasks, their nature as parametric methods is associated with several inevitable drawbacks: they are insufficient in modeling uncertainties; they are vulnerable to overfitting, especially on small datasets; the learning is difficult and their success relies heavily on the choices of network structures and hyperparameters. A natural question to be asked is that can we analyze graph-structured data with non-parametric models such as Gaussian processes? A few recent publications show us that the answer is positive [18–20]. However, the investigations up to now are far from thorough and complete, and the development of deep Gaussian processes has also revealed us an alternative approach that is scarcely explored by far.

In this paper we propose Stochastic **D**eep **G**aussian **P**rocesses over **G**raphs (DGPG), which is a method for modeling the relations between input and output signals over graph-structured domains. Our work is closely related to [11]: both our methods are based on sparse approximation [10] and the variational inference framework [21]. Though the evidence lower bound is analytically intractable, it can be evaluated and optimized by the stochastic minibatch sampling technique.

We summarize the main contributions of this paper as follows: 1) We propose a novel Bayesian non-parametric method called Stochastic Deep Gaussian Processes over Graphs (DGPG), to model the relations of input/output signals over graphs; 2) It is rigorously proved that under some technical assumptions, the sampling variances of DGPG are strictly less then that of [11], implying that DGPG achieves faster convergence by considering graph information; 3) We performed experiments on a synthetic dataset. Numerical results conform with our theoretical analysis and support the claim that DGPG converges much faster and better by utilizing graph information; 4) Experiments on realistic datasets demonstrate that DGPG can be successfully applied to both small and large-scale datasets. We show that our method outperforms a recent GP-based graph learning algorithm, and is competitive to a state-of-the-art DNN method on the challenging task of traffic flow prediction; 5) We show that DGPG possesses several other desirable characteristics: it can model uncertainties with high accuracy, and the automatic relevance determination (ARD) kernel allows it to learn which neighbors and features are of greater importance for the prediction.

## 2 Background and Related Work

### 2.1 Sparse Gaussian Processes

Gaussian processes (GPs) [1] are among the most popular choices in the Bayesian non-parametric machine learning family. Given a dataset $\mathcal{D} = \{(\mathbf{x}_i, y_i)\}_{i=1}^N$ with $\mathbf{x}_i \in \mathbb{R}^D$ and $y_i \in \mathbb{R}$, we denote $\mathbf{X} = (\mathbf{x}_1, ..., \mathbf{x}_N)^\top$, $\mathbf{y} = (y_1, ..., y_N)^\top$, and the latent function values as $\mathbf{f} = f(\mathbf{X})$. Gaussian process regression (GPR) models the relation as a function $f : \mathbb{R}^D \to \mathbb{R}$, assuming $f$ has a Gaussian prior. The prediction has a closed form when the likelihood $p(\mathbf{y}|\mathbf{f})$ is Gaussian. The prior of $f$ is characterized by a kernel function $k : \mathbb{R}^D \times \mathbb{R}^D \to \mathbb{R}$ and a mean function $m : \mathbb{R}^D \to \mathbb{R}$. For a test point $\mathbf{x}^*$, its prediction $y^*$ also follows the Gaussian distribution and is analytically tractable. Standard GPs possess various advantages, however the prohibitive $O(N^3)$ training complexity renders them inapplicable to large-scale datasets. Recently lots of work emerged to assuage the complexity burden [22–26], among which the sparse Gaussian processes stand out as some of the most successful and popular methods [27–32].

Sparse Gaussian processes (SGPs) introduce $M$ inducing inputs $\mathbf{Z} = (\mathbf{z}_1, ..., \mathbf{z}_M)^\top$, and the corresponding function values are $\mathbf{u} = f(\mathbf{Z})$ where $p(\mathbf{u}) = \mathcal{N}(\mathbf{u}|m(\mathbf{Z}), k(\mathbf{Z}, \mathbf{Z}))$. The joint probability density is $p(\mathbf{y}, \mathbf{f}, \mathbf{u}) = p(\mathbf{f}|\mathbf{u}; \mathbf{X}, \mathbf{Z})p(\mathbf{u}; \mathbf{Z}) \prod_{i=1}^N p(y_i|\mathbf{f}_i)$. Posterior $p(\mathbf{f}, \mathbf{u}|\mathbf{y})$ is only tractable when the likelihood $p(y_i|\mathbf{f}_i)$ is Gaussian, and in this case it still requires $O(N^3)$ computation. Fortunately, variational inference presents us a framework to solve these two problems simultaneously. Variational inference introduces a distribution $q(\mathbf{f}, \mathbf{u})$ to approximate the posterior $p(\mathbf{f}, \mathbf{u}|\mathbf{y})$. It can be showed that minimizing the Kullback-Leibler divergence $\mathrm{KL}(q(\mathbf{f}.\mathbf{u})||p(\mathbf{f}, \mathbf{u}|\mathbf{y}))$ is equivalent to maximizing the evidence lower bound (ELBO), which is defined as $\mathcal{L}_{SGP} = \mathbb{E}_{q(\mathbf{f}, \mathbf{u})} \left[\log \frac{p(\mathbf{y}, \mathbf{f}, \mathbf{u})}{q(\mathbf{f}, \mathbf{u})}\right]$.

As in [30] the variational distribution is defined as $q(\mathbf{f}, \mathbf{u}) = p(\mathbf{f}|\mathbf{u}; \mathbf{X}, \mathbf{Z})q(\mathbf{u})$ where $q(\mathbf{u}) = N(\mathbf{u}|\mathbf{m}, \mathbf{S})$ is assumed to be Gaussian. Furthermore, the conditional distribution $p(\mathbf{f}|\mathbf{u}; \mathbf{X}, \mathbf{Z})$ is also Gaussian: $p(\mathbf{f}|\mathbf{u}; \mathbf{X}, \mathbf{Z}) = N(\mathbf{f}|\boldsymbol{\mu}, \boldsymbol{\Sigma})$, where $[\boldsymbol{\mu}]_i = m(\mathbf{x}_i) + \boldsymbol{\alpha}(\mathbf{x}_i)^\top (\mathbf{u} - m(\mathbf{Z}))$, $[\boldsymbol{\Sigma}]_{ij} = k(\mathbf{x}_i, \mathbf{x}_j) - \boldsymbol{\alpha}(\mathbf{x}_i)^\top k(\mathbf{Z}, \mathbf{Z})\boldsymbol{\alpha}(\mathbf{x}_j)$, and $\boldsymbol{\alpha}(\mathbf{x}_i) = k(\mathbf{Z}, \mathbf{Z})^{-1}k(\mathbf{Z}, \mathbf{x}_i)$. Finally we can show that

the marginal also follows Gaussian: $q(\mathbf{f}|\mathbf{m},\mathbf{S};\mathbf{X},\mathbf{Z}) = \int p(\mathbf{f}|\mathbf{u};\mathbf{X},\mathbf{Z})q(\mathbf{u})d\mathbf{u} = \mathcal{N}(\mathbf{f}|\tilde{\boldsymbol{\mu}},\tilde{\boldsymbol{\Sigma}})$, where

$$[\tilde{\boldsymbol{\mu}}]_i = \mu_{\mathbf{m},\mathbf{Z}}(\mathbf{x}_i) = m(\mathbf{x}_i) + \boldsymbol{\alpha}(\mathbf{x}_i)^\top (\mathbf{m} - m(\mathbf{Z})), \tag{1}$$

$$[\tilde{\boldsymbol{\Sigma}}]_{ij} = \Sigma_{\mathbf{S},\mathbf{Z}}(\mathbf{x}_i,\mathbf{x}_j) = k(\mathbf{x}_i,\mathbf{x}_j) - \boldsymbol{\alpha}(\mathbf{x}_i)^\top (k(\mathbf{Z},\mathbf{Z}) - \mathbf{S})\boldsymbol{\alpha}(\mathbf{x}_j). \tag{2}$$

With the analysis above, ELBO $\mathcal{L}_{SGP}$ can be further simplified, so that by maximizing it both the variational parameters and the latent variables can be found.

## 2.2 Doubly Stochastic Deep Gaussian Process

In [11] the authors show that it is possible to stack the above single-layer models and form a hierarchical deep Gaussian process (DGP). For a DGP model with $L$ layers, it introduces $L$ independent latent variables $\{\mathbf{U}^l\}_{l=1}^L$ as the function values of the inducing points $\{\mathbf{Z}^l\}_{l=1}^L$, so that $p(\mathbf{Y},\{\mathbf{F}^l,\mathbf{U}^l\}_{l=1}^L) = \prod_{i=1}^N p(\mathbf{y}_i|\mathbf{f}_i^L)\prod_{l=1}^L p(\mathbf{F}^l|\mathbf{U}^l;\mathbf{F}^{l-1},\mathbf{Z}^{l-1})p(\mathbf{U}^l;\mathbf{Z}^{l-1})$. With a similar reasoning by assuming $q(\mathbf{U}^l) = \mathcal{N}(\mathbf{U}^l|\mathbf{m}^l,\mathbf{S}^l)$, we have $q(\{\mathbf{F}^l\}_{l=1}^L) = \prod_{l=1}^L \mathcal{N}(\mathbf{F}^l|\tilde{\boldsymbol{\mu}}^l,\tilde{\boldsymbol{\Sigma}}^l)$, where $[\tilde{\boldsymbol{\mu}}^l]_i = \mu_{\mathbf{m}^l,\mathbf{Z}^{l-1}}(\mathbf{f}_i^l)$ and $[\tilde{\boldsymbol{\Sigma}}^l]_{ij} = \Sigma_{\mathbf{S}^l,\mathbf{Z}^{l-1}}(\mathbf{f}_i^l,\mathbf{f}_j^l)$ ($\mu_{\mathbf{m}^l,\mathbf{Z}^{l-1}}$ and $\Sigma_{\mathbf{S}^l,\mathbf{Z}^{l-1}}$ defined in Eq (1) (2)).

A key finding of [11] is that for a given $\mathbf{x}_i$, the marginal of the variational distribution in the last layer $q(\mathbf{f}_i^L)$ depends only on the marginals of $\{q(\mathbf{f}_i^l)\}_{l=1}^{L-1}$ in the previous layers. Formally we have $q(\mathbf{f}_i^L) = \int \prod_{l=1}^{L-1} q(\mathbf{f}_i^l|\mathbf{m}^l,\mathbf{S}^l;\mathbf{f}_i^{l-1},\mathbf{Z}^{l-1}) df_i^l$. Consequently $\mathbf{f}^l$ depends only on $\mathbf{f}^{l-1}$, so that the sampling of $\mathbf{f}_i^L$ in the last layer can be conducted in a recursive layer-to-layer manner by using the "re-parameterization' trick [33, 34]. After introducing the hierarchical structure, it can be showed that the new ELBO is $\mathcal{L}_{DGP} = \sum_{i=1}^N \mathbb{E}_{q(\mathbf{f}_i^L)}\left[\log p(\mathbf{y}_n|\mathbf{f}_n^L)\right] - \sum_{l=1}^L \mathrm{KL}\left[q(\mathbf{U}^l)\|p(\mathbf{U}^l;\mathbf{Z}^{l-1})\right]$.

With the samples of $\mathbf{f}_i^L$ in the last layer readily available, [11] finally proposed to evaluate and optimize ELBO with stochastic sampling. Similar to SGP, DGP can be applied to large-scale datasets (e.g. a billion data points), and the deep structure allows it to model more complex relations. Later we will see that our new method enhances this recursive sampling scheme by utilizing graph information, and our sampling has strictly smaller variances so that better and faster convergence can be achieved. All the aforementioned advantages of SGP and DGP are naturally inherited by our method.

## 2.3 Related Work

To assuage the $O(N^3)$ training complexity of standard GPs, several sparse approximation methods have been proposed [35, 36, 27–29, 31, 32], the ideas of which are to construct a manageable set of inducing points to summarize the information of the original dataset. It is rigorously established in [37] that under some technical assumptions sparse approximation methods can produce reliable results with $M = O(\log^D N)$ inducing points, where $D$ is the dimension of the data. Inspired by some of them, [10] introduces a method to select the inducing points by using variational inference. A few years later this work has become the foundation of a series of deep Gaussian process models: inspired by [10] and the GP latent variable model (GP-LVM) [38], Damianou and Lawrence proposed deep GPs [7], which are composed of hierarchical Gaussian process mappings; several other deep GP models were lately proposed [8, 9]. All these models demonstrate their superior performances over a variety of challenging tasks. The work above derives analytically tractable evidence lower bounds, while in contrast with them in [11] - which is the most related paper of our work - the authors present a model with milder independent assumption and utilize the sampling technique. Applying the sampling technique not only allows the model to use both Gaussian (e.g. in regression) and non-Gaussian (e.g. in classification) likelihood in an uniform manner, but also enables it to take advantage of GPU acceleration. These strengths are also inherited by our proposal.

Recently applying machine learning methods to graph-structured datasets has draw much attention from the research community. Among various approaches, graph neural networks (GNNs) [39, 40] are one family of the most successful methods. GNNs can be divided into two categories, the spectral methods [12, 15, 16] and the non-spectral methods [13, 14, 17]. Though with great empirical success in many challenging tasks including network analysis, biochemistry, and traffic prediction - to name only a few of them - their training is generally difficult and the success relies heavily on the choices of network structures and hyperparameters, and the models are vulnerable to overfitting. It is thus desirable that non-parametric methods like GPs can overcome these drawbacks.

A few works aiming to apply GPs to graph-structured domains exist. The paper of [18] proposes a GP model to solve semi-supervised learning problems over graphs. Their method is also based on sparse approximation and the variational inference framework. However, our goals are different and our approaches share little similarity besides the minor points mentioned above. In [19] the authors present a novel model called the graph convolutional GPs. Their paper aims to tackle the graph classification problem by designing an addictive kernel with translation-invariant property, which borrows the idea of the convolutional operator on the grid-like domains. Gaussian Markov Random Fields (GMRFs) [41, 42] are elegant models for interpolating signals over graphs. However they are inadequate in extrapolation across the vertices, which is the challenge that focused in our work. Finally a method called Gaussian Processes over Graphs (GPG) was proposed in [20]. This paper shares the same goal with us, i.e. both our models try to learn the mappings between input-output signals over graphs. However our methodologies are totally different. The key idea of their approach is to incorporate the graph information as prior knowledge in the covariance matrix. Their method still suffers the poor scalability, and results are only reported for datasets up to about 100 data points. By contrast DGPG is based on sparse approximation and deep GPs. Consequently our method can handle datasets with tens of thousands of training instances, and has the potential of modeling more complex mappings.

## 3    Stochastic Deep Gaussian Processes over Graphs

**Problem Statement and Notations**  The dataset is defined as $\mathcal{D} = \{G, \Psi, \Phi\}$, where $G$ encodes the graph structure, and $\Psi$ and $\Phi$ represent the input and output signals. To be more precise, $G = \langle \mathcal{V}, \mathcal{E} \rangle$ is comprised of a set of vertices $\mathcal{V}$ and a set of edged $\mathcal{E} \subseteq \mathcal{V} \times \mathcal{V}$. For each $v_i \in \mathcal{V}$, we use $pa(v_i) = \{v_k | (v_k, v_i) \in \mathcal{E}\}$ to denote its parents. Each input $\psi \in \Psi$ is a function $\psi : \mathcal{V} \to \mathbb{R}^{d_{in}}$ over the vertices of the graph. Similarly each output $\phi \in \Phi$ is defined as $\phi : \mathcal{V} \to \mathbb{R}^{d_{out}}$. The goal is to learn a function $h : \Psi \to \Phi$ that takes a signal $\psi$ as input and predicts the output $\phi$.

We introduce several notations that makes the presentation more succinct. Let $N$ be the size of the training dataset, $|\mathcal{V}| = K$ be the number of vertices. The graph $G$ is denoted by $A_G \in \{0, 1\}^{K \times K}$. Note that binary connection does not restrict the expressiveness, since the ARD kernel can capture the relevance of each node (see later discussions). The input signals $\Psi$ can be represented as a matrix $\mathbf{X} = (\mathbf{x}_1, ..., \mathbf{x}_N)^\top$, where $\mathbf{x}_i \in \mathbb{R}^{K d_{in}}$ is a vector constructed by concatenating the features of all the nodes, depicted as the $i$-th row of $\mathbf{X}$ in Fig. 1. Similarly $\Phi$ is represented as $\mathbf{Y} = (\mathbf{y}_1, ..., \mathbf{y}_N)^\top$, where $\mathbf{y}_i \in \mathbb{R}^{K d_{out}}$ is depicted as the $i$-th row of $\mathbf{Y}$ in Fig. 1.

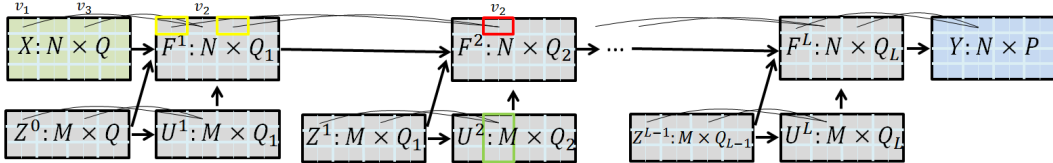

**Figure 1:** An example illustrating the notations and factorization. $\mathbf{X}$ and $\mathbf{Y}$ are the observed input and output signals; $Q_l = K d_l$ is the dimension of the concatenated vector in layer $l$. In this example $v_2$ is connected to $v_1$ and $v_3$; two yellow boxes denote $\mathbf{F}_1^{1,1}$ and $\mathbf{F}_1^{1,3}$ (or $\mathbf{F}_1^{1,pa(2)}$); red box denotes $\mathbf{F}_1^{2,2}$; green box denotes $\mathbf{U}^{2,2}$.

**Model Structure**  Similar to [11], we define a deep structure in our DGPG. For a $L$-layer model, we denote $\mathbf{F}^1, ..., \mathbf{F}^L$ ($\mathbf{F}^l \in \mathbb{R}^{N \times K d^l}$ where $d^l$ is the dimension of each node in layer $l$) as the latent function values. Inspired by the sparse approximation framework, for each layer we introduce $M$ inducing points $\mathbf{Z}^1, ..., \mathbf{Z}^L$ ($\mathbf{Z}^l \in \mathbb{R}^{M \times K d^l}$) and the associated function values are $\mathbf{U}^1, ..., \mathbf{U}^L$ ($\mathbf{U}^l \in \mathbb{R}^{M \times K d^l}$). For a matrix $\mathbf{M}$ ($\mathbf{M} \in \{\mathbf{X}, \mathbf{Y}, \mathbf{F}, \mathbf{Z}, \mathbf{U}\}$), we use $\mathbf{M}^{l,k}$ to denote the submatrix associated with the $l$-th layer and the $k$-th vertex. $\mathbf{M}^{l,pa(k)}$ is the submatrix in the $l$-th layer associated with all the parents of vertex $v_k$ in the next layer. $\mathbf{M}_i^{l,k}$ is a vector that denotes the $i$-th row of $\mathbf{M}^{l,k}$.

The factorization of the joint distribution is depicted in Fig. 1, given by

$$p\left(\mathbf{Y}, \{\mathbf{F}^{l,k}, \mathbf{U}^{l,k}\}_{l,k}\right) = \underbrace{\prod_{n=1}^{N} \prod_{k=1}^{K} p\left(\mathbf{y}_n^k | \mathbf{F}_n^{L,k}\right)}_{\text{likelihood}} \cdot$$

$$\underbrace{\prod_{l=1}^{L} p\left(\mathbf{F}^{l,k} | \mathbf{U}^{l,k}; \mathbf{F}^{l-1,pa(k)}, \mathbf{Z}^{l-1,pa(k)}\right) p\left(\mathbf{U}^{l,k}; \mathbf{Z}^{l-1,pa(k)}\right)}_{\text{GP prior}}, \quad (3)$$

where $\mathbf{F}^0 = \mathbf{X}$. It is assumed that the prior is Gaussian, and $\mathbf{U}^{l,k}$'s are factorized among layers and vertices. The likelihood can be Gaussian (e.g. regression) or non-Gaussian (e.g. classification).

By applying variational inference, we define the approximate posterior distribution as

$$q(\{\mathbf{F}^{l,k}, \mathbf{U}^{l,k}\}_{l,k}) = \prod_{l=1}^{L} \prod_{k=1}^{K} p(\mathbf{F}^{l,k} | \mathbf{U}^{l,k}; \mathbf{F}^{l-1,pa(k)}, \mathbf{Z}^{l-1,pa(k)}) q(\mathbf{U}^{l,k}), \quad (4)$$

where $q(\mathbf{U}^{l,k}) \doteq \mathcal{N}(\mathbf{U}^{l,k} | \mathbf{m}^{l,k}, \mathbf{S}^{l,k})$ is Gaussian. Since both two terms on the r.h.s. in Eq (4) are Gaussian, we can marginalize out $\mathbf{U}^{l,k}$ and obtain $q(\{\mathbf{F}^{l,k}\}_{l,k}) = \prod_{l=1}^{L} \prod_{k=1}^{K} \mathcal{N}(\mathbf{F}^{l,k} | \tilde{\boldsymbol{\mu}}^{l,k}, \tilde{\boldsymbol{\Sigma}}^{l,k})$, where $[\tilde{\boldsymbol{\mu}}^{l,k}]_i = \mu_{\mathbf{m}^{l,k}, \mathbf{Z}^{l-1,pa(k)}}(\mathbf{F}_i^{l-1,pa(k)})$, $[\tilde{\boldsymbol{\Sigma}}]_{ij} = \Sigma_{\mathbf{S}^{l,k}, \mathbf{Z}^{l-1,pa(k)}}(\mathbf{F}_i^{l-1,pa(k)}, \mathbf{F}_j^{l-1,pa(k)})$. In this equation $[\tilde{\boldsymbol{\mu}}^{l,k}]_i$ denotes the $i$-th row of the matrix $\tilde{\boldsymbol{\mu}}^{l,k} \in \mathbb{R}^{N \times Kd^l}$; $[\tilde{\boldsymbol{\Sigma}}^{l,k}]_{ij}$ denotes the value at the entry $(i,j)$ in $\tilde{\boldsymbol{\Sigma}}^{l,k} \in \mathbb{R}^{N \times N}$. The final analytical results are can be obtained by Eq (1) and (2). Considering the above mean and covariance, as well as the marginal property of Gaussian processes, we have the following simple method to sample the latent function values recursively.

**Recursive Sampling Scheme** The marginal of the $i$-th row, $k$-th vertex in layer $L$ depends only on the parents of $k$ in layer $L-1$. By a recursive analysis, the distribution of $q(\mathbf{F}_i^{L,k})$ depends only on the ancestors of $k$. Consequently for the final layer we have

$$q(\mathbf{F}_i^{L,k}) = \int \prod q(\mathbf{F}_i^{l,k'} | \mathbf{m}^{l,k'}, \mathbf{S}^{l,k'}; \mathbf{F}_i^{l-1,pa(k')}, \mathbf{Z}^{l-1,pa(k')}) d\{\mathbf{F}_i^{l-1,pa(k')}\}, \quad (5)$$

where $\langle l, k' \rangle$'s enumerate all **ancestors** of $\langle L, k \rangle$ and $\langle L, k \rangle$ itself.

The significance of the above observation is that it provides us with a method to sample from $q(\mathbf{F}^{L,k})$ in a recursive way. Similar to [11], the sampling can be conveniently achieved by using the "reparameterization trick" recursively. Specifically let $\boldsymbol{\epsilon}_i^{l,k} \sim \mathcal{N}(0, \mathbf{I}_{d^l})$ be a sampled Gaussian noise, the samples of vertices in different layers can be obtained recursively as

$$\hat{\mathbf{F}}_i^{l,k} = \mu_{\mathbf{m}^{l,k}, \mathbf{Z}^{l-1,pa(k)}}(\hat{\mathbf{F}}_i^{l-1,pa(k)}) + \boldsymbol{\epsilon}_i^{l,k} \odot \sqrt{\Sigma_{\mathbf{S}^{l,k}, \mathbf{Z}^{l-1,pa(k)}}(\hat{\mathbf{F}}_i^{l-1,pa(k)}, \hat{\mathbf{F}}_i^{l-1,pa(k)})} \quad (6)$$

**Evidence Lower Bound** Now we derive the evidence lower bound of DGPG. As $\mathbf{F}^{l,k}$'s and $\mathbf{U}^{l,k}$'s represent the latent variables in the model, the evidence lower bound is computed as $\mathcal{L}_{DGPG} = \mathbb{E}_{q(\{\mathbf{F}^{l,k}, \mathbf{U}^{l,k}\}_{l,k})} \left[ \log \frac{p(\mathbf{Y}, \{\mathbf{F}^{l,k}, \mathbf{U}^{l,k}\}_{l,k})}{q(\{\mathbf{F}^{l,k}, \mathbf{U}^{l,k}\}_{l,k})} \right]$. By substituting Eq (3) and (4) this can be simplified as

$$\mathcal{L}_{DGPG} = \sum_{n=1}^{N} \sum_{k=1}^{K} \mathbb{E}_{q(\mathbf{F}_n^{L,k})} \left[ \log p\left(\mathbf{y}_n^k | \mathbf{F}_n^{L,k}\right) \right] - \sum_{l=1}^{L} \sum_{k=1}^{K} \text{KL} \left[ q\left(\mathbf{U}^{l,k}\right) \| p\left(\mathbf{U}^{l,k}; \mathbf{Z}^{l-1,pa(k)}\right) \right].$$

There all totally $O(KNM^2(d^1 + \cdots + d^L))$ variational parameters in the model.

**Main Theorem** Next we present the main theoretical result of this paper, which states that by utilizing graph information DGPG reduces sampling variances, which implies faster convergence during the optimization of ELBO. First we list two technical assumptions and their intuitive explanations.

Assumption 1. $k(\mathbf{Z}, \mathbf{Z}) - \mathbf{S} \succ 0$ ($\succ 0$ denotes positive definite). Intuitively in Eq (2) the uncertainty characterized by $\Sigma_{\mathbf{S}, \mathbf{Z}}(\mathbf{x}_i, \mathbf{x}_j)$ should be strictly smaller than that of $k(\mathbf{x}_i, \mathbf{x}_j)$ since more information is available. Similar observations and arguments can be found in [27].

Assumption 2. Without graph the prediction $y$ depends on all other nodes, i.e. $h(\mathbf{x}) = y$ where $\mathbf{x} \in \mathbb{R}^D$; while with graph $y$ depends only on $\tilde{\mathbf{x}} \subset \mathbf{x}$, i.e. $h(\tilde{\mathbf{x}}) = y$ where $\tilde{\mathbf{x}} \in \mathbb{R}^d$ and $d < D$. Similarly

$\tilde{\mathbf{Z}} \in \mathbb{R}^{M \times d}$ and $\tilde{\mathbf{S}} \in \mathbb{R}^{M \times M}$ represent the variables when graph is considered. We assume that $||k(\tilde{\mathbf{Z}}, \tilde{\mathbf{x}})||^2 / ||k(\mathbf{Z}, \mathbf{x})||^2 > \lambda_{max}/\tilde{\lambda}_{min}$ ($\lambda_{max}$ is the maximal eigenvalue of $k(\mathbf{Z}, \mathbf{Z})^{-1}(k(\mathbf{Z}, \mathbf{Z}) - \mathbf{S})k(\mathbf{Z}, \mathbf{Z})^{-1}$, and $\tilde{\lambda}_{min}$ is the minimal eigenvalue of $k(\tilde{\mathbf{Z}}, \tilde{\mathbf{Z}})^{-1}(k(\tilde{\mathbf{Z}}, \tilde{\mathbf{Z}}) - \tilde{\mathbf{S}})k(\tilde{\mathbf{Z}}, \tilde{\mathbf{Z}})^{-1}$). In the proof we show how to justify this assumption when the graph is sparse (i.e. $d \ll D$).

**Theorem 1** *We denote the sampling variance with graph as $\tilde{\mathbf{\Sigma}}_{ii} = \Sigma_{\tilde{\mathbf{S}}, \tilde{\mathbf{Z}}}(\tilde{\mathbf{x}}_i, \tilde{\mathbf{x}}_i)$ (defined in Eq (2)), and the sampling variance without graph as $\mathbf{\Sigma}_{ii} = \Sigma_{\mathbf{S}, \mathbf{Z}}(\mathbf{x}_i, \mathbf{x}_i)$. Under the assumptions 1 and 2, for a large variety of kernel functions (including RBF, Matérn32, etc.) we have $\tilde{\mathbf{\Sigma}}_{ii} < \mathbf{\Sigma}_{ii}$.*

The proof is left in the supplementary materials. Finally let $\alpha = ||k(\tilde{\mathbf{Z}}, \tilde{\mathbf{x}})||^2 / ||k(\mathbf{Z}, \mathbf{x})||^2$ and $\beta = \lambda_{max}/\tilde{\lambda}_{min}$. The assumptions are equivalent to $\lambda_{min}(\tilde{\lambda}_{min}) > 0$ and $\alpha > \beta$. Now both the assumptions become verifiable assertions, which will be further validated in the experimental study.

## 4 Experimental Study

In this section, we conduct thorough experimental study of our method. Our implementation is based on the GPflow framework [43] and the implementation of DGP [3]. The source code and evaluations of our work are publicly available [4]. All experiments were performed on a Ubuntu server equipped with GeForce GTX 1080 Ti GPU, which supports our parallelized implementation.

### 4.1 Synthetic Dataset

We begin with presenting the experimental results on a synthetic dataset. The goal of this example is to support our theoretical analysis with numerical evidences, e.g. verifying the positive definite assumption and the claim that graph can reduce sampling variances and lead to faster convergence.

Consider a symmetric graph with 500 nodes, and each node is connected to approximately 7 randomly selected parents (including a self-connecting edge). We sample each input signal $\mathbf{x}_i \in \mathbb{R}^{500}$ from a standard multivariate normal distribution with an isometric covariance matrix $\mathbf{x} \sim \mathcal{N}(0, I)$. The output of each node is defined as the sum of its parents: $\mathbf{y}_i^k = \sum_{k': k' \in pa(k)} \mathbf{x}_i^{k'}$. We totally generate 500 input-output signals for training, and 200 for testing. In the theoretical analysis we show that $\alpha \gg \beta$ implies the algorithm benefits from graph information and thus converges faster, so we use $r = \alpha/\beta$ to measure the richness of information of graphs: larger $r$ suggests more informative graph structures. We further corrupt the graph by randomly select $p = 0, 2, 4, 6$ edges and change them into erroneous connections (when $p = 0$ the connection is not changed).

**Table 1:** Several statistics that are of interest for the theoretical analysis.

| Run | $p = 0$ | | | | | | $p = 2$ | $p = 4$ | $p = 6$ |
|-----|---------------|---------------|--------------------|--------|--------|---------------|---------------|---------------|---------------|
| | $\lambda_{min}$ | $\lambda_{max}$ | $\tilde{\lambda}_{min}$ | $\alpha$ | $\beta$ | $\alpha/\beta$ | $\alpha/\beta$ | $\alpha/\beta$ | $\alpha/\beta$ |
| 1 | 0.4 | 1.1 | 1.1E+00 | 5.0E+18 | 1E+0 | 5.0E+18 | 2.9E+17 | 2.8E+15 | 6.1E+13 |
| 2 | 1.2 | 1.2 | 5.4E+02 | 4.1E+17 | 2E-3 | 2.1E+20 | 2.8E+16 | 3.3E+15 | 4.2E+13 |
| 3 | 1.0 | 1.1 | 4.9E+01 | 2.8E+17 | 2E-2 | 1.4E+19 | 1.1E+16 | 6.8E+14 | 4.5E+13 |
| 4 | 0.6 | 1.0 | 2.8E+00 | 3.7E+20 | 3E-1 | 1.2E+21 | 3.9E+15 | 3.3E+15 | 3.5E+13 |
| 5 | 0.4 | 1.1 | 9.5E-01 | 1.3E+18 | 4E-1 | 3.3E+18 | 6.0E+16 | 1.8E+16 | 1.6E+13 |

We trained a DGPG model with 1 layer and 50 inducing points, repeated over 5 runs. Without loss of generality we report the results of the first node and the first instance. Table 1 presents various numerical values that are of interest to our analysis. As $p$ increases $r$ becomes smaller (corrupted graph is less informative), which follows our analysis and expectation. Fig. 2 shows the convergences ELBO. 'DGPG without graph' is computed with a fully connected graph, in which case DGPG reduces to DGP.

The results suggest that: 1) Assumption $k(\mathbf{Z}, \mathbf{Z}) - \mathbf{S} \succ 0$ does always hold ($\lambda_{min}$'s are always positive); 2) $\alpha \gg \beta$ - according to Theorem 1 this implies that our method can reduce the sampling variances by utilizing the graph information; 3) the above claim is further demonstrated in Fig. 3, from which we can see that our method converges much faster in optimizing ELBO.

## 4.2 Small Datasets

Data efficiency is an appealing property of GPs. In this experiment we test the performance of DGPG on three small datasets: Weather [44], fMRI [45] and ETEX [46]. To ensure that our comparison is fair, we use the identical datasets and training/test splitting. Description of the datasets is presented in Table 2. We use $M = 5$ inducing points. When building the adjacent matrix we use Euclidean distance to construct the connections.

Considering the utilization of graph information, we add two graph-aware Gaussian Process methods (GPG [20] and GCGP [19]) to the comparison experiment. Accordingly, other DGP methods are excluded as they are incapable of modeling graph datasets. GCGP [19] is originally designed to only make one output for the whole graph

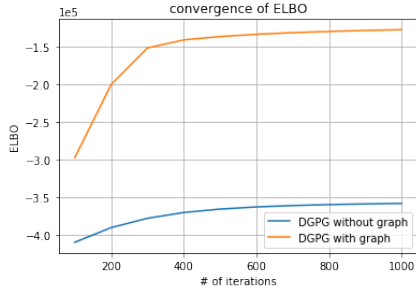

**Figure 2:** Convergence of ELBO

**Table 2:** Dataset description of training/test instances, number of nodes, and average degree. #Iter. is the steps of iteration.

| Dataset | Weather | fMRI | ETEX |
|---|---|---|---|
| #Training | 46 | 145 | 30 |
| #Test | 46 | 145 | 30 |
| #Node | 45 | 100 | 168 |
| Avg. D. | 5 | 5 | 10 |
| #Iter. | 2000 | 5000 | 5000 |

rather than every node. In order to make it comparable, we train an independent GCGP instance for each node and recompose their outputs as the requested complete prediction. Besides, the definition of intrinsic angular distance on general graphs is not provided in [19], so we just use the diffusion graph convolution kernel in experiments.

Table 3 presents the evaluation results. We are interested in three metrics: mean absolute error (MAE), root mean square error (RMSE) and mean absolute percentage error (MAPE). Since these metrics were not considered in [20] we evaluated their method locally. We also consider two naive baselines: historical mean and historical median. In the preprocessing stage we normalize the data on each node to have zero mean and one standard deviation. For fMRI and ETEX the input signals on some nodes are absent, and we simply take their values as Gaussian noises with very small variances ($10^{-4}$).

We tested our model with 1 to 4 layers, and tried the RBF and Matérn32 kernels. From this experiment we can draw the following conclusions: 1) our method outperforms the other recently proposed Gaussian-based method in most cases, and sometimes by a large margin; 2) in complex tasks deeper layers lead to better performance, showing that DGPG benefits from the deep structure; 3) RBF and Matérn32 kernels produce similar results, indicating that DGPG is not sensitive to the particular kernel choice; 4) DGPG reserves the appealing data efficiency property.

**Table 3:** GP-L and GPG-L are baselines in [20]. SVR denotes support vector regression. For SVR the output is a function of its neighbors' input and high-order graph information is lost. Metrics of GCGP [19] are calculated using the recomposed prediction result. We report the results of DGPG using linear kernel (L), RBF kernel (RBF), Matérn32 kernel (M32), and the optimal layer. Terms with underline denote best results. Results of several other baselines are in the supplementary materials.

| Dataset | Metrics | mean | GP-L | GPG-L | SVR | GCGP | DGPG (L/RBF/M32/layer) |
|---|---|---|---|---|---|---|---|
| Weather | MAE | 1.61 | 1.52 | 1.66 | 1.44 | 1.46 | 1.47 / 1.37 / 1.36 / 1 |
| | RMSE | 2.11 | 1.97 | 2.19 | 1.88 | 1.90 | 1.92 / 1.80 / 1.79 / 1 |
| | MAPE | 17% | 19% | 24% | 15% | 18% | 17% / 16% / 15% / 1 |
| fMRI | MAE | 0.016 | 0.026 | 0.074 | 0.014 | 0.015 | 0.015 / 0.010 / 0.010 / 4 |
| | RMSE | 0.021 | 0.033 | 0.089 | 0.020 | 0.021 | 0.020 / 0.015 / 0.015 / 4 |
| | MAPE | 1.6% | 2.6% | 7.4% | 1.4% | 1.5% | 1.5% / 1.0% / 1.0% / 4 |
| ETEX | MAE | 0.40 | 0.25 | 0.30 | 0.18 | 0.22 | 0.30 / 0.27 / 0.21 / 3 |
| | RMSE | 0.45 | 0.34 | 0.36 | 0.31 | 0.31 | 0.41 / 0.41 / 0.35 / 3 |
| | MAPE | 40% | 25% | 30% | 18% | 22% | 30% / 27% / 21% / 3 |

**Table 5:** Comparison on the task of traffic flow prediction. Results of other baselines are obtained from [48]. DGPG* utilizes validation data during training, fixed to be 3-layer. Terms with underline indicate best results. Terms with wavy underline indicate second best. Results of the BAY dataset are in the supplementary materials.

| | T | Metrics | VAR | FC-LSTM | DCRNN | DGPG (1/2/3/4) | DGPG* |
|---|---|---|---|---|---|---|---|
| LA | 15 min | MAE | 4.42 | 3.44 | 2.77 | 3.06 / 3.04 / 3.02 / 3.02 | 3.00 |
| | | RMSE | 7.89 | 6.30 | 5.38 | 5.40 / 5.35 / 5.32 / 5.32 | 5.31 |
| | | MAPE | 10.2% | 10.9% | 7.3% | 6.6% / 6.0% / 6.6% / 6.5% | 6.5% |
| | 30 min | MAE | 5.41 | 3.77 | 3.15 | 3.57 / 3.42 / 3.42 / 3.39 | 3.39 |
| | | RMSE | 9.13 | 7.23 | 6.45 | 6.37 / 6.16 / 6.16 / 6.12 | 6.13 |
| | | MAPE | 12.7% | 10.9 | 8.8% | 7.5% / 7.3% / 7.3% / 7.2% | 7.2% |
| | 60 min | MAE | 6.52 | 4.37 | 3.6 | 4.02 / 3.83 / 3.00 / 3.80 | 3.8 |
| | | RMSE | 10.11 | 8.69 | 7.59 | 7.12 / 6.93 / 6.94 / 6.94 | 6.85 |
| | | MAPE | 15.8% | 13.2% | 10.5% | 8.4% / 8.1% / 7.9% / 8.0% | 5.0% |

## 4.3 Large Datasets

Next we test DGPG on two challenging large datasets, where the goal is to predict the traffic flow with different forecasting horizons (15, 30 and 60 min) in Los Angeles (LA) [47] and Bay Area (BAY) [48]. We compare with DCRNN [48], which is one of state-of-the-art approaches in traffic flow prediction. We use the same data and splitting according to their released code[5]. We use $M = 20$ inducing points per node. Records in some instances are missing, and we fill them with the median values. When building the graph each vertex is connected to about 5 neighbors. The only difference is that we only consider the first 100 nodes. Dataset details can be found in the supplementary materials.

The results are presented in Table 5. Due to space limit we only show three strong baselines, VAR [49], FC-LSTM [50] and DCRNN (other results are in the supplementary materials). One advantage of DGPG is that we do not have strong necessity to use a validation set, so similar to [18] we utilize the validation dataset for training while fixing its layer to be 3. These results are showed as DGPG*.

Results in Table 5 show that: 1) DGPG is competitive w.r.t. the state-of-the-art method DCRNN, and outperforms it in most cases in the more challenging dataset LA; 2) DGPG produces accurate predictions for different datasets and forecasting horizons, showing its stability and consistency; 3) DGPG achieves its best performance with the appropriate layers, demonstrating that it benefits from deep structures; 4) the performance can be improved by utilizing the validation data during training.

**Variance Analysis** A distinguishing advantage of GPs is that they are capable of modeling the predictive variances. We will see that DGPG can capture uncertainty with satisfactory precision.

Inspired by [51], we exam how many test instances fall in the predictive confidence interval. For Gaussian distribution $\mathcal{N}(\mu, \sigma^2)$, the interval $(\mu - k\sigma, \mu + k\sigma)$ covers about 68.3%/95.5%/99.7% of the probability density as $k = 1, 2, 3$. So we can expect that the potion of the test instances falling in the predictive intervals should have the same ratio. Table 4 shows the experimental results. We can see that the numerical results comply with our analysis reasonably well, particularly within the $\pm 2\sigma$ intervals.

**Table 4:** Variance Analysis of DGPG

| | T | $\pm 1\sigma$ | $\pm 2\sigma$ | $\pm 3\sigma$ |
|---|---|---|---|---|
| LA | 15 min | 84.4% | 94.7% | 97.7% |
| | 30 min | 84.2% | 94.5% | 97.5% |
| | 60 min | 83.7% | 94.3% | 97.5% |
| BAYS | 15 min | 87.4% | 95.6% | 97.9% |
| | 30 min | 86.0% | 94.5% | 97.2% |
| | 60 min | 85.1% | 94.1% | 97.0% |

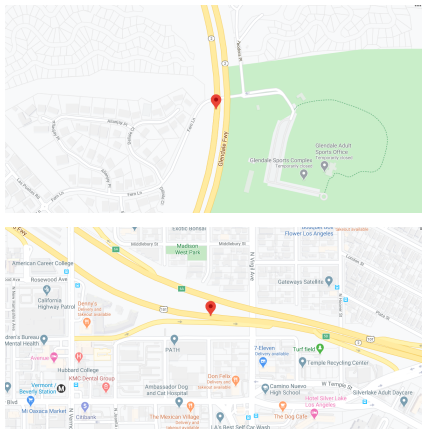

**Figure 3:** Locations of the sensors with: (top) the least uncertainty, and (bottom) the largest uncertainty.

We performed a case study to show DGPG's ability in capturing uncertainty. We computed the variances of the test data and took the average across each node. Fig. 3 depicts the two sensors with the "least" and the "largest" uncertainties. The sensor with the least uncertainty locates at sparsely populated area with simple traffic condition; the sensor with the largest uncertainty locates at the business center where the traffic condition is very complex.

**Automatic Relevance Determination** Fig. 4 depicts the learned lengthscales of node 1. In this particular example, node 1 is connected to 5 other nodes (including itself). Each node has 8 features: 'T' represents time in a day (00:00, 00:05, ...); 'D' represents day in a week (Monday, Tuesday, etc.); features at $t = \{6, ..., 1\}$ represent the traffic at $5t$ minutes before. The first 8 bars correspond to the lengthscales of the self connection, and other bars represent the lengthscales w.r.t. the other four nodes. Note that a smaller lengthscale indicates greater relevance.

The effectiveness of the ARD kernel can be inspected in Fig. 4: 1) by using ARD, DGPG is capable of discerning which neighbors are more relevant - for instance the self connection is more important when the forecasting horizon is small (15 min); 2) DGPG can further discover which features are relevant - "time" plays a significant role in the prediction, while which "day" the record occurred is less relevant; 3) we emphasize again that the relevance is automatically discovered, so that using binary adjacent matrices does not reduce the expressiveness of our model.

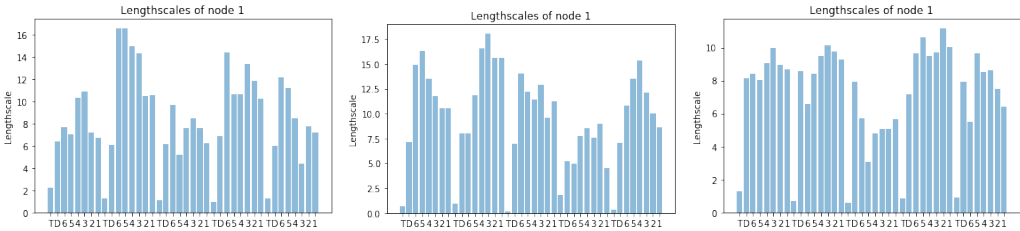

**Figure 4:** Lengthscales inferred by the ARD kernel in: (left) 15 min; (middle) 30 min; (right) 60 min.

## 5 Conclusion

In this paper we propose a method to learn the mappings between input and output signals over graphs, which we call Stochastic Deep Gaussian Processes over Graphs (DGPG). Our method utilizes the graph information during the construction of the deep GP structure. By applying variational inference, the evidence lower bound is derived and optimized by a recursive sampling scheme.

We conducted thorough experiments in both synthetic and realistic datasets. Numerical results on a synthetic dataset validate our theoretical assumptions and analysis, particularly the claim that our method has smaller sampling variances and thus converges faster. In the realistic datasets, we show that our method generally outperforms other baselines in small datasets, and is competitive to the state-of-the-art method in the challenging task of traffic flow prediction. Finally DGPG also exhibits several appealing characteristics, such as the ability to accurately model uncertainties and to automatically discover which vertices and features are relevant to the prediction.

## Acknowledgments and Disclosure of Funding

This work is supported in part by the National Key Research and Development Program of China under Grant 2018YFB1800204, the National Natural Science Foundation of China under Grant 61771273, the R&D Program of Shenzhen under Grant JCYJ20180508152204044, and the project "PCL Future Greater-Bay Area Network Facilities for Large-scale Experiments and Applications (LZC0019)".

## Broader Impact

This paper proposes a graph guided deep Gaussian process model. Through incorporating the graph information into deep Gaussian process, our proposed method reduces sampling variances and achieves faster convergence. Graph structure is a very common tool in numerous fields. In experiments, we also show that the proposed method can facilitate the tasks of traffic flow prediction. We believe that the exploration of graph structure will largely improve the capacity of current machine learning algorithms.

## Footnotes

[3]https://github.com/ICL-SML/Doubly-Stochastic-DGP

[4]https://github.com/naiqili/DGPG

[5]https://github.com/liyaguang/DCRNN

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
