[Supplementary Material]

# Supplementary Materials

**Naiqi Li**[1,*]   **Wenjie Li**[2,*]   **Jifeng Sun**[1]   **Yinghua Gao**[2]   **Yong Jiang**[1,3]   **Shu-Tao Xia**[2,3]

[1]Tsinghua-Berkeley Shenzhen Institute, Tsinghua University
[2]Shenzhen International Graduate School, Tsinghua University
[3]PCL Research Center of Networks and Communications, Peng Cheng Laboratory

## 1   Proof of the Main Theorem

We prove the main theorem by assuming the RBF kernel, though the reasoning can be directly applied to a variety of other popular kernels, including Matérn32, Matérn52, exponential, and rational quadratic kernel functions. We list the assumptions below for convenience.

Assumption 1. $k(\mathbf{Z}, \mathbf{Z}) - \mathbf{S} \succ 0$ ($\succ 0$ denotes positive definite).

Assumption 2. Without graph the prediction $y$ depends on all other nodes, i.e. $h(\mathbf{x}) = y$ where $\mathbf{x} \in \mathbb{R}^D$; while with graph $y$ depends only on $\tilde{\mathbf{x}} \subset \mathbf{x}$, i.e. $h(\tilde{\mathbf{x}}) = y$ where $\tilde{\mathbf{x}} \in \mathbb{R}^d$ and $d < D$. Similarly $\tilde{\mathbf{Z}} \in \mathbb{R}^{M \times d}$ and $\tilde{\mathbf{S}} \in \mathbb{R}^{M \times M}$ represent the variables when graph is considered. We assume that $||k(\tilde{\mathbf{Z}}, \tilde{\mathbf{x}})||^2/||k(\mathbf{Z}, \mathbf{x})||^2 > \lambda_{max}/\tilde{\lambda}_{min}$ ($\lambda_{max}$ is the maximal eigenvalue of $k(\mathbf{Z}, \mathbf{Z})^{-1}(k(\mathbf{Z}, \mathbf{Z}) - \mathbf{S})k(\mathbf{Z}, \mathbf{Z})^{-1}$, and $\tilde{\lambda}_{min}$ is the minimal eigenvalue of $k(\tilde{\mathbf{Z}}, \tilde{\mathbf{Z}})^{-1}(k(\tilde{\mathbf{Z}}, \tilde{\mathbf{Z}}) - \tilde{\mathbf{S}})k(\tilde{\mathbf{Z}}, \tilde{\mathbf{Z}})^{-1}$).

**Remark 1.** *We show that assumption 2 is justified when the graph is sparse, i.e. $d \ll D$.*

*Justification.* By definition we have

$$k(\mathbf{Z}, \mathbf{x}) = (k(\mathbf{z}_1, \mathbf{x}_1), ..., k(\mathbf{z}_M, \mathbf{x}_M))^\top, \tag{1}$$

$$k(\tilde{\mathbf{Z}}, \tilde{\mathbf{x}}) = (k(\tilde{\mathbf{z}}_1, \tilde{\mathbf{x}}_1), ..., k(\tilde{\mathbf{z}}_M, \tilde{\mathbf{x}}_M))^\top. \tag{2}$$

Our approach is to firstly show that for all $j$, $k(\tilde{\mathbf{z}}_j, \tilde{\mathbf{x}}) \gg k(\mathbf{z}_j, \mathbf{x})$.

Without loss of generalization, assume that $\tilde{\mathbf{z}}_j \in \mathbb{R}^d$ corresponds to the first $d$ dimensions in $\mathbf{z}_j \in \mathbb{R}^D$. Similarly, $\tilde{\mathbf{x}} \in \mathbb{R}^d$ corresponds to the first $d$ dimensions in $\mathbf{x} \in \mathbb{R}^D$. In other words, with graph information the prediction depends only on the first $d$ dimensions of all the $D$ feature.

It is easy to see that when $d \ll D$,

$$||\mathbf{x} - \mathbf{z}||^2 = \sum_{i=1}^{d} (x_i - z_i)^2 + \sum_{i=d+1}^{D} (x_i - z_i)^2 \tag{3}$$

$$\gg \sum_{i=1}^{d} (\tilde{x}_i - \tilde{z}_i)^2 \tag{4}$$

$$= ||\tilde{\mathbf{x}} - \tilde{\mathbf{z}}||^2 \tag{5}$$

Thus we have

$$k(\tilde{\mathbf{x}}, \tilde{\mathbf{z}}) = \sigma^2 \exp\{-\frac{1}{\lambda}||\tilde{\mathbf{x}} - \tilde{\mathbf{z}}||^2\} \gg \sigma^2 \exp\{-\frac{1}{\lambda}||\mathbf{x} - \mathbf{z}||^2\} = k(\mathbf{x}, \mathbf{z}). \tag{6}$$

---

[*]Equal contributions.

Finally since for all $j$ we have $k(\tilde{\mathbf{z}}_j, \tilde{\mathbf{x}}) \gg k(\mathbf{z}_j, \mathbf{x})$, and the RBF kernel function is positive, we have $||k(\tilde{\mathbf{Z}}, \tilde{\mathbf{x}})|| \gg ||k(\mathbf{Z}, \mathbf{x})||$. Particularly we require that $||k(\tilde{\mathbf{Z}}, \tilde{\mathbf{x}})||^2 / ||k(\mathbf{Z}, \mathbf{x})||^2 > \lambda_{max} / \tilde{\lambda}_{min}$.

Rigorously, step (4) further requires that $|x_i - z_i|$ is lower bounded by some constant term, and in the last step $\lambda_{max} / \tilde{\lambda}_{min}$ need to be upper bounded by some constant. We omit these details since they are not essential for the following proofs, and the assumption has been empirically verified. $\square$

The sampling variance is calculated by:

$$\Sigma_{\mathbf{S}, \mathbf{Z}}(\mathbf{x}_i, \mathbf{x}_j) = k(\mathbf{x}_i, \mathbf{x}_j) - \boldsymbol{\alpha}(\mathbf{x}_i)^\top (k(\mathbf{Z}, \mathbf{Z}) - \mathbf{S})\boldsymbol{\alpha}(\mathbf{x}_j), \tag{7}$$

$$\boldsymbol{\alpha}(\mathbf{x}_i) = k(\mathbf{Z}, \mathbf{Z})^{-1} k(\mathbf{Z}, \mathbf{x}_i). \tag{8}$$

We first prove a simple lemma.

**Lemma 1.** *Let* $\mathbf{M} = k(\mathbf{Z}, \mathbf{Z})^{-1}(k(\mathbf{Z}, \mathbf{Z}) - \mathbf{S})k(\mathbf{Z}, \mathbf{Z})^{-1}$. $\mathbf{M}$ *is positive definite (*$\mathbf{M} \succ 0$*).*

*Proof.* According to the property of kernel function, the kernel matrix $k(\mathbf{Z}, \mathbf{Z})$ is positive definite, so $k(\mathbf{Z}, \mathbf{Z})^{-1} \succ 0$. Also note that $k(\mathbf{Z}, \mathbf{Z})^{-1}$ is symmetry. For any $\mathbf{x} \neq 0$, we have

$$\mathbf{x}^\top k(\mathbf{Z}, \mathbf{Z})^{-1}(k(\mathbf{Z}, \mathbf{Z}) - \mathbf{S})k(\mathbf{Z}, \mathbf{Z})^{-1}\mathbf{x} \tag{9}$$

$$= \mathbf{x}^\top (k(\mathbf{Z}, \mathbf{Z})^{-1})^\top (k(\mathbf{Z}, \mathbf{Z}) - \mathbf{S})k(\mathbf{Z}, \mathbf{Z})^{-1}\mathbf{x} \tag{10}$$

$$= (k(\mathbf{Z}, \mathbf{Z})^{-1}\mathbf{x})^\top (k(\mathbf{Z}, \mathbf{Z}) - \mathbf{S})k(\mathbf{Z}, \mathbf{Z})^{-1}\mathbf{x}. \tag{11}$$

We know $k(\mathbf{Z}, \mathbf{Z}) - \mathbf{S} \succ 0$ by assumption 1, and $k(\mathbf{Z}, \mathbf{Z})^{-1}\mathbf{x} \neq \mathbf{0}$ (since $k(\mathbf{Z}, \mathbf{Z})^{-1} \succ 0$, $\mathbf{x} \neq 0$), so $\mathbf{x}^\top k(\mathbf{Z}, \mathbf{Z})^{-1}(k(\mathbf{Z}, \mathbf{Z}) - \mathbf{S})k(\mathbf{Z}, \mathbf{Z})^{-1}\mathbf{x} > 0$ always holds. This implies $\mathbf{M} \succ 0$.

$\square$

**Theorem 1.** *We denote the sampling variance with graph as* $\tilde{\boldsymbol{\Sigma}}_{ii} = \Sigma_{\tilde{\mathbf{S}}, \tilde{\mathbf{Z}}}(\tilde{\mathbf{x}}_i, \tilde{\mathbf{x}}_i)$ *(defined in Eq (7)), and the sampling variance without graph as* $\boldsymbol{\Sigma}_{ii} = \Sigma_{\mathbf{S}, \mathbf{Z}}(\mathbf{x}_i, \mathbf{x}_i)$. *Under the assumptions 1 and 2, for a large variety of kernel functions (including RBF, Matérn32 kernels, etc.) we have* $\tilde{\boldsymbol{\Sigma}}_{ii} < \boldsymbol{\Sigma}_{ii}$.

*Proof.* As in Lemma 1, we define

$$\mathbf{M} = k(\mathbf{Z}, \mathbf{Z})^{-1}(k(\mathbf{Z}, \mathbf{Z}) - \mathbf{S})k(\mathbf{Z}, \mathbf{Z})^{-1}, \tag{12}$$

$$\tilde{\mathbf{M}} = k(\tilde{\mathbf{Z}}, \tilde{\mathbf{Z}})^{-1}(k(\tilde{\mathbf{Z}}, \tilde{\mathbf{Z}}) - \tilde{\mathbf{S}})k(\tilde{\mathbf{Z}}, \tilde{\mathbf{Z}})^{-1}. \tag{13}$$

According to Lemma 1, both $\mathbf{M}$ and $\tilde{\mathbf{M}}$ are positive definite.

From fundamental linear algebra we know that the Rayleigh quotient, which is defined as $\frac{\mathbf{v}^\top \mathbf{A} \mathbf{v}}{\mathbf{v}^\top \mathbf{v}}$, lies within the smallest and the largest eigenvalues of $\mathbf{A}$ for all $\mathbf{v}$. So we have

$$\lambda_{min} \leq \frac{k(\mathbf{Z}, \mathbf{x}_i)^\top \mathbf{M} k(\mathbf{Z}, \mathbf{x}_i)}{k(\mathbf{Z}, \mathbf{x}_i)^\top k(\mathbf{Z}, \mathbf{x}_i)} \leq \lambda_{max}, \tag{14}$$

$$\tilde{\lambda}_{min} \leq \frac{k(\tilde{\mathbf{Z}}, \tilde{\mathbf{x}}_i)^\top \tilde{\mathbf{M}} k(\tilde{\mathbf{Z}}, \tilde{\mathbf{x}}_i)}{k(\tilde{\mathbf{Z}}, \tilde{\mathbf{x}}_i)^\top k(\tilde{\mathbf{Z}}, \tilde{\mathbf{x}}_i)} \leq \tilde{\lambda}_{max}. \tag{15}$$

Consequently, we have

$$k(\tilde{\mathbf{Z}}, \tilde{\mathbf{x}}_i)^\top \tilde{\mathbf{M}} k(\tilde{\mathbf{Z}}, \tilde{\mathbf{x}}_i) \geq \tilde{\lambda}_{min} ||k(\tilde{\mathbf{Z}}, \tilde{\mathbf{x}}_i)||^2 \qquad \text{(by Eq (15))} \tag{16}$$

$$> \tilde{\lambda}_{min} \cdot \frac{\lambda_{max}}{\tilde{\lambda}_{min}} ||k(\mathbf{Z}, \mathbf{x}_i)||^2 \qquad \text{(by Assumption 2)} \tag{17}$$

$$\geq k(\mathbf{Z}, \mathbf{x}_i)^\top \mathbf{M} k(\mathbf{Z}, \mathbf{x}_i) \qquad \text{(by Eq (14))} \tag{18}$$

Substituting Eq (8) (12) (13) into the inequality, it can be showed that

$$\boldsymbol{\alpha}(\tilde{\mathbf{x}}_i)^\top (k(\tilde{\mathbf{Z}}, \tilde{\mathbf{Z}}) - \tilde{\mathbf{S}}) \boldsymbol{\alpha}(\tilde{\mathbf{x}}_i) \tag{19}$$

$$=k(\tilde{\mathbf{Z}}, \tilde{\mathbf{x}}_i)^\top k(\tilde{\mathbf{Z}}, \tilde{\mathbf{Z}})^{-1} (k(\tilde{\mathbf{Z}}, \tilde{\mathbf{Z}}) - \tilde{\mathbf{S}}) k(\tilde{\mathbf{Z}}, \tilde{\mathbf{Z}})^{-1} k(\tilde{\mathbf{Z}}, \tilde{\mathbf{x}}_i) \tag{20}$$

$$=k(\tilde{\mathbf{Z}}, \tilde{\mathbf{x}}_i)^\top \tilde{\mathbf{M}} k(\tilde{\mathbf{Z}}, \tilde{\mathbf{x}}_i) \tag{21}$$

$$>k(\mathbf{Z}, \mathbf{x}_i)^\top \mathbf{M} k(\mathbf{Z}, \mathbf{x}_i) \tag{22}$$

$$=k(\mathbf{Z}, \mathbf{x}_i)^\top k(\mathbf{Z}, \mathbf{Z})^{-1} (k(\mathbf{Z}, \mathbf{Z}) - \mathbf{S}) k(\mathbf{Z}, \mathbf{Z})^{-1} k(\mathbf{Z}, \mathbf{x}_i) \tag{23}$$

$$=\boldsymbol{\alpha}(\mathbf{x}_i)^\top (k(\mathbf{Z}, \mathbf{Z}) - \mathbf{S}) \boldsymbol{\alpha}(\mathbf{x}_i). \tag{24}$$

Noting that for the RBF kernel, which is defined as $k(\mathbf{x}_i, \mathbf{x}_j) = \sigma^2 \exp\{-\frac{1}{\lambda} \|\mathbf{x}_i - \mathbf{x}_j\|^2\}$, we have $k(\mathbf{x}_i, \mathbf{x}_i) = k(\tilde{\mathbf{x}}_i, \tilde{\mathbf{x}}_i) = \sigma^2$.

Finally we have

$$\tilde{\boldsymbol{\Sigma}}_{ii} = k(\tilde{\mathbf{x}}_i, \tilde{\mathbf{x}}_i) - \boldsymbol{\alpha}(\tilde{\mathbf{x}}_i)^\top (k(\tilde{\mathbf{Z}}, \tilde{\mathbf{Z}}) - \tilde{\mathbf{S}}) \boldsymbol{\alpha}(\tilde{\mathbf{x}}_i) \tag{25}$$

$$= \sigma^2 - \boldsymbol{\alpha}(\tilde{\mathbf{x}}_i)^\top (k(\tilde{\mathbf{Z}}, \tilde{\mathbf{Z}}) - \tilde{\mathbf{S}}) \boldsymbol{\alpha}(\tilde{\mathbf{x}}_i) \tag{26}$$

$$< k(\mathbf{x}_i, \mathbf{x}_i) - \boldsymbol{\alpha}(\mathbf{x}_i)^\top (k(\mathbf{Z}, \mathbf{Z}) - \mathbf{S}) \boldsymbol{\alpha}(\mathbf{x}_i) \tag{27}$$

$$= \boldsymbol{\Sigma}_{ii} \tag{28}$$

This completes the proof. □

## 2 Experimental Details

### 2.1 Synthetic Dataset

Consider a symmetric graph with 500 nodes, and each node is connected to approximately 7 randomly selected parents (including a self-connecting edge). We sample each input signal $\mathbf{x}_i \in \mathbb{R}^{500}$ from a standard multivariate normal distribution with an isometric covariance matrix $\mathbf{x} \sim \mathcal{N}(0, I)$ (i.e. every two dimensions are independent). The output of each node is defined as the sum of its parents: $\mathbf{y}_i^k = \sum_{k':k' \in pa(k)} \mathbf{x}_i^{k'}$. We totally generate 500 input-output signals for training, and 200 for testing.

Figure 1: (a) Convergence of ELBO; (b) Convergence of the likelihood variance.

### 2.2 Small Dataset

#### 2.2.1 Weather Dataset

This dataset includes the temperature data of 45 cities in Sweden during October to December 2017 [1]. The goal is to use the data of the past 5 day to forecast tomorrow's temperature.

**Table 1:** Several statistics that are of interest to the theoretical analysis. $||\mathbf{k}||$ denotes $||k(\mathbf{Z}, \mathbf{x})||$; $||\tilde{\mathbf{k}}||$ denotes $||k(\tilde{\mathbf{Z}}, \tilde{\mathbf{x}})||$; $\lambda_{min}$ ($\lambda_{max}$) is the minimal (maximal) eigenvalue of $k(\mathbf{Z}, \mathbf{Z})^{-1}(k(\mathbf{Z}, \mathbf{Z}) - \mathbf{S})k(\mathbf{Z}, \mathbf{Z})^{-1}$; $\tilde{\lambda}_{min}$ ($\tilde{\lambda}_{max}$) is the minimal (maximal) eigenvalue of $k(\tilde{\mathbf{Z}}, \tilde{\mathbf{Z}})^{-1}(k(\tilde{\mathbf{Z}}, \tilde{\mathbf{Z}}) - \tilde{\mathbf{S}})k(\tilde{\mathbf{Z}}, \tilde{\mathbf{Z}})^{-1}$; $\alpha = ||k(\tilde{\mathbf{Z}}, \tilde{\mathbf{x}})||^2 / ||k(\mathbf{Z}, \mathbf{x})||^2$ and $\beta = \lambda_{max} / \tilde{\lambda}_{min}$.

| Run | $||\mathbf{k}||$ | $||\tilde{\mathbf{k}}||$ | $\lambda_{min}$ | $\lambda_{max}$ | $\tilde{\lambda}_{min}$ | $\tilde{\lambda}_{max}$ | $\alpha$ | $\beta$ |
|-----|---------|--------|--------|--------|---------|---------|----------|----------|
| 1 | 2.3E-09 | 5.2 | 0.4 | 1.1 | 1.1E+00 | 1.8E+07 | 5.0E+18 | 1.1E+00 |
| 2 | 7.1E-09 | 4.6 | 1.2 | 1.2 | 5.4E+02 | 8.9E+06 | 4.1E+17 | 2.3E-03 |
| 3 | 9.0E-09 | 4.7 | 1.0 | 1.1 | 4.9E+01 | 2.6E+07 | 2.8E+17 | 2.3E-02 |
| 4 | 3.0E-10 | 5.8 | 0.6 | 1.0 | 2.8E+00 | 7.9E+07 | 3.7E+20 | 3.7E-01 |
| 5 | 4.6E-09 | 5.2 | 0.4 | 1.1 | 9.5E-01 | 1.4E+07 | 1.3E+18 | 4.1E-01 |

**Table 2:** Dataset description, presenting the number of training/test instances, number of nodes, and average degree of the graph.

| Dataset | Weather [1] | fMRI [2] | ETEX [3] |
|---------|-------------|----------|----------|
| #Training | 46 | 145 | 30 |
| #Test | 46 | 145 | 30 |
| #Node | 45 | 100 | 168 |
| Avg. Degree | 5 | 5 | 10 |

### 2.2.2 fMRI Dataset

This dataset contains the functional magnetic resonance image (fMRI) data of the cerebellum region of brain [2]. The task is to predict the activation levels of other 90 vertices according to the input signals on 10 vertices.

### 2.2.3 ETEX Dataset

This contains the atmospheric tracer measurements recorded in the European Tracer Experiment [3]. The task is to forecast the tracer concentration recorded by 84 ground stations according to the data from the other 88 locations.

## 2.3 Large Dataset

### 2.3.1 LA Dataset

This dataset contains traffic information collected by the sensors on the highway of Los Angeles countries during Mar 1st 2012 to Jun 30th 2012 [4]. Totally 23974 instances are used for training, 3425 for validation and 6850 for testing.

### 2.3.2 BAY Dataset

This dataset contains traffic information of the Bay Area collected by California Transportation Agencies (CalTrans) Performance Measurement System (PeMS), during Jan 1st 2017 to May 31st 2017 [5]. Totally 36465 instances are used for training, 5209 for validation and 10419 for testing.

**Table 3:** Comparison with another Gaussian-based approach. Mean and median indicate historical mean and historical median baselines. GP-L/K and GPG-L/K are baselines in [6]. SVR/RT/MLP denote support vector regression/regression tree/Multilayer perceptron respectively. For these three regression methods the output is a function of its neighbors' input and high-order graph information is lost. GCGP [7] makes one output for the whole graph, so we predict each node separately when performing experiments with it. We report the results of DGPG using linear kernel (L), RBF kernel (RBF), Matérn32 kernel (M32), and the optimal layer. Terms with underline denote best results.

| Dataset | Metrics | mean | median | GP-L | GPG-L | GP-K | GPG-K | SVR | RT | MLP | GCGP | DGPG (L/RBF/M32/layer) |
|---|---|---|---|---|---|---|---|---|---|---|---|---|
| Weather | MAE | 1.61 | 1.67 | 1.52 | 1.66 | 3.06 | 3.15 | 1.44 | 1.77 | 1.60 | 1.46 | 1.47 / 1.37 / 1.36 / 1 |
|  | RMSE | 2.11 | 2.23 | 1.97 | 2.19 | 3.85 | 3.94 | 1.88 | 2.35 | 2.06 | 1.90 | 1.92 / 1.80 / 1.79 / 1 |
|  | MAPE | 17% | 18% | 19% | 24% | 37% | 39% | 15% | 23% | 21% | 18% | 17% / 16% / 15% / 1 |
| fMRI | MAE | 0.016 | 0.015 | 0.026 | 0.074 | 0.022 | 0.078 | 0.014 | 0.014 | 0.016 | 0.015 | 0.015 / 0.010 / 0.010 / 4 |
|  | RMSE | 0.021 | 0.022 | 0.033 | 0.089 | 0.028 | 0.093 | 0.020 | 0.021 | 0.024 | 0.021 | 0.020 / 0.015 / 0.015 / 4 |
|  | MAPE | 1.6% | 1.5% | 2.6% | 7.4% | 2.2% | 7.8% | 1.4% | 1.4% | 1.6% | 1.5% | 1.5% / 1.0% / 1.0% / 4 |
| ETEX | MAE | 0.40 | 0.35 | 0.25 | 0.30 | 0.38 | 0.40 | 0.18 | 0.14 | 0.17 | 0.22 | 0.30 / 0.27 / 0.21 / 3 |
|  | RMSE | 0.45 | 0.50 | 0.34 | 0.36 | 0.48 | 0.48 | 0.31 | 0.31 | 0.31 | 0.31 | 0.41 / 0.41 / 0.35 / 3 |
|  | MAPE | 40% | 36% | 25% | 30% | 38% | 40% | 18% | 14% | 17% | 22% | 30% / 27% / 21% / 3 |

**Table 4:** Comparison on the task of traffic flow prediction. Results of other baselines are obtained from [5]. DGPG* utilizes validation data during training, fixed to be 3-layer. Terms with underline indicate best results. Terms with wavy underline indicate second best. Results of other baselines are taken from paper [5].

| | T | Metrics | HA | ARIMA | VAR | SVR | FNN | FC-LSTM | DCRNN | DGPG (1/2/3/4) | DGPG* |
|---|---|---|---|---|---|---|---|---|---|---|---|
| LA | 15 min | MAE | 4.16 | 3.99 | 4.42 | 3.99 | 3.99 | 3.44 | 2.77 | 3.06 / 3.04 / 3.02 / 3.02 | 3.00 |
| | | RMSE | 7.80 | 8.21 | 7.89 | 8.45 | 7.94 | 6.30 | 5.38 | 5.40 / 5.35 / 5.32 / 5.32 | 5.31 |
| | | MAPE | 13.0% | 9.6% | 10.2% | 9.3% | 9.9% | 9.6% | 7.3% | 6.6% / 6.0% / 6.6% / 6.5% | 6.5% |
| | 30 min | MAE | 4.16 | 5.15 | 5.41 | 5.05 | 4.23 | 3.77 | 3.15 | 3.57 / 3.42 / 3.42 / 3.39 | 3.39 |
| | | RMSE | 7.80 | 10.45 | 9.13 | 10.87 | 8.17 | 7.23 | 6.45 | 6.37 / 6.16 / 6.16 / 6.12 | 6.13 |
| | | MAPE | 13.0% | 12.7% | 12.7% | 12.1% | 12.9% | 10.9% | 8.8% | 7.5% / 7.3% / 7.3% / 7.2% | 7.2% |
| | 60 min | MAE | 4.16 | 6.90 | 6.52 | 6.72 | 4.49 | 4.37 | 3.60 | 4.02 / 3.83 / 3.00 / 3.80 | 3.80 |
| | | RMSE | 7.80 | 13.23 | 10.11 | 13.76 | 8.69 | 8.69 | 7.59 | 7.12 / 6.93 / 6.94 / 6.94 | 6.85 |
| | | MAPE | 13.0% | 17.4% | 15.8% | 16.7% | 14.0% | 13.2% | 10.5% | 8.4% / 8.1% / 7.9% / 8.0% | 5.0% |
| BAY | 15 min | MAE | 2.88 | 1.62 | 1.74 | 1.85 | 2.20 | 2.05 | 1.38 | 1.64 / 1.67 / 1.75 / 1.79 | 1.71 |
| | | RMSE | 5.59 | 3.30 | 3.16 | 3.59 | 4.42 | 4.19 | 2.95 | 3.44 / 3.37 / 3.46 / 3.49 | 3.25 |
| | | MAPE | 6.8% | 3.5% | 3.6% | 3.8% | 5.2% | 4.8% | 2.9% | 3.3% / 3.4% / 3.6% / 3.6% | 3.5% |
| | 30 min | MAE | 2.88 | 2.33 | 2.32 | 2.48 | 2.30 | 2.20 | 1.74 | 2.19 / 2.17 / 2.12 / 2.51 | 2.08 |
| | | RMSE | 5.59 | 4.76 | 4.25 | 5.18 | 4.63 | 4.55 | 3.97 | 4.55 / 4.37 / 4.36 / 5.06 | 4.23 |
| | | MAPE | 6.8% | 5.4% | 5.0% | 5.5% | 5.4% | 5.2% | 3.9% | 4.3% / 4.3% / 4.3% / 4.8% | 4.2% |
| | 60 min | MAE | 2.88 | 3.38 | 2.93 | 3.28 | 2.46 | 2.37 | 2.07 | 2.66 / 2.46 / 2.50 / 2.56 | 2.44 |
| | | RMSE | 5.59 | 6.50 | 5.44 | 7.08 | 4.98 | 4.96 | 4.74 | 5.28 / 5.08 / 5.06 / 5.25 | 4.99 |
| | | MAPE | 6.8% | 8.3% | 6.5% | 8.0% | 5.9% | 5.7% | 4.9% | 5.1% / 4.9% / 4.9% / 5.0% | 4.8% |