[Reviews · NeurIPS 2020]

Review 1

Summary and Contributions: The authors propose using doubly stochastic variational inference to compute the relationship between the input and output signals on the vertices of a graph under a Gaussian process prior.

Strengths: The method appears to be novel & potentially quite powerful. It allows 'side information' provided by the graph structure to be introduced into the model. Good selection of empirical experiments. Interesting proof for the variance scale. Finally it may improve on the rather fragile optimisation that complex models in this field can suffer from.

Weaknesses: One thing that worries me a lot is that all the evaluations are about the posterior mean: How well does it handle the posterior (co)variance? Does it properly reach across the relevant part of the domain or is the posterior distribution very concentrated? VI suffers from this anyway, I imagine that a graph structure like this might compound the problem.

Correctness: To my knowledge this seems valid. The empirical assessment seems correct from what we are told. I'm not sure about the proof for the variance. Do the numbers in Table 2 & 3 have units? I'm not sure what cross-validation split was conducted for the experiments?

Clarity: I think it would benefit from a simple example and/or some clarity about what the task is (in words, before getting to the problem statement's maths? What are the input/output for example wasn't immediately clear.

Relation to Prior Work: I found the background to be thorough (to the best of my knowledge) & puts the paper in context.

Reproducibility: Yes

Additional Feedback: It's unclear if the inducing points were optimised? Also how long (how many iterations) did these models need? (fig 2 shows this for one example). ln 218 - "In total, we generate" might be better than "We totally..." ==After discussions and reading the author feedback== My assessment was pulled in two directions: At first, having read the other reviews & feedback, I was going to downgrade my assessment as I've read a bit more background and think the application of DGP over graphs is a relatively straightforward step in terms of the surrounding literature. But not enough to make me 'reject' but not feel strongly about the accept. But then I read the author's point that they did look at uncertainty, reread that section and feel that actually my criticism was largely unwarranted on that count. So my overall score will remain unchanged (although I'm not going to increase my confidence score, as I don't feel I'm an expert in this area).


Review 2

Summary and Contributions: This paper introduces a deep GP model that makes predictions on nodes of a graph. It takes advantage of the graph structure to create a purposefully constrained model to help make predictions. The paper provides an empirical evaluation on several noisy datasets with graph structure, with impressive

Strengths: The paper develops the model based on the well-developed theoretical principles of variational inference in GP models. By applying these techniques to a sensible model, the authors achieved good performance. The model is generally applicable to many regression on tasks to grids, so it is likely to be useful for many researchers and practitioners in the community.

Weaknesses: - Limitations of theorem 1 The theorem on the variance of the predictions (theorem 1) is a bit odd, and it is unclear how much value it really adds. The theorem effectively tries to prove that the predictive variance of the approximate posterior is smaller for the graph-aware model than for a plain DGP. This depends only on the covariance of the inducing point distribution q(u). This covariance is learned by optimising the ELBO, and so it is hard to prove anything about it short of simply running the training procedure. Assumption 2 doesn't provide much additional clarity, as it effectively places a constraint on \tilde{S} as a function of S. This assumption also cannot be verified without running the algorithm. In short, it is not clear that Theorem 1 makes any statements about the algorithm that cannot be made simply by empirically keeping track of the predictive variance while running the algorithm. In addition, the empirical evaluation isn't making a fair comparison between the two models. The empirical evaluation claims that the graph-aware DGP converges faster than the normal DGP. This is not at all obvious from figure 2. If anything, it seems (as expected) that the normal DGP simply converges to a much lower value, which is explained by it not being a suitable model. - Test on simpler models I wonder to what extent the non-linearity of the GPs are really needed, given the small number of inducing points that are used. For the small datasets M = 5. For such small numbers of inducing points, GPs often behave like linear models. It would be good to include an experiment using a linear kernel to see how much the non-linearity of the GP really helps. In addition, it would be good to include a short discussion on how the number of inducing points was chosen. Also, how were the inducing points initialized?

Correctness: The methodology and empirical evaluation are broadly correct.

Clarity: The paper is broadly clearly written, with a few spots of lack of clarity. - §2.2: The description of a DGP as a probabilistic model is very short, and could do with more details. - §3: The connectivity structure of what nodes in the graph a GP depends on is the single key contribution of this paper. It would be good to have an illustration showing a graph, and what nodes are used as input to the next-layer GP. - §3: The recursive sampling scheme is presented in an unclear way, to the point of being incorrect. A better way to present this would be simply to explicitly describe the recursive procedure for obtaining a sample from the function values at the last layer. Equation 5 does not correctly represent the correlations of function evaluations between layers. A confusing aspect of Salimbeni's DGP formulation (which is confusingly presented in the orignal paper), is that there is a distinction between the approximate GPs for each layer, which are independent, and the function evaluations between layers, which are dependent. See e.g. Thang Bui's thesis (http://mlg.eng.cam.ac.uk/thang/docs/papers/thesis-thang.pdf) page 94.

Relation to Prior Work: Prior work is well discussed. 2 comments: - It would be helpful for a reader to have a discussion on why this graph-based structure was chosen. Were there any approaches in the literature that directly inspired the choices that were made? The structure of the proposed method does look a lot like those from the Graph Convolutional Network literature. Although these papers are cited, if these were an inspiration, it would be good to have a more explicit discussion. - Around line 72 there is a discussion of sparse methods helping with the complexity burden of GPs. [1] provides proofs for regression that these approximations are very accurate with only small numbers of inducing points. [1] http://proceedings.mlr.press/v97/burt19a.html

Reproducibility: Yes

Additional Feedback: Overall, the paper presents a sensible approach to an important problem, with impressive empirical results. While I recommend accepting, I hope that my issues will be addressed in the rebuttal so I can raise my score.


Review 3

Summary and Contributions: The paper modifies a Deep Gaussian Process to operate on a graph (input and outputs on a Graph). Inference proceeds by a modification of the doubly stochastic VI framework by Salimbeni & Deisenroth, which is modified to consider the neighbourhood of a vertex/node when computing the marginal at layer l leading to a recursive sampling scheme. There are a few synthetic, small and large real world datasets being used based on prior work [20,45] from Graph NN and shallow GP approaches on Graphs. The paper offers a main theorem (similar to [20]) that the DGPG has a reduced sampling variance (compared to a vanilla DGP) because it exploits the Markov property/graph structure.

Strengths: The paper does examine multiple datasets and compares to a shallow GP [20] on a small dataset and with Graph ANNs on the large dataset. The approach seems technically correct and it is an improvement over ignoring the graph structure and naively running a DGP. The standard ARD kernel in this graph setting naturally allows (given the parents from the graph structure) to learn relative importance between these parents and also relative importance between features in each node.

Weaknesses: I think the novelty is somewhat limited and summarised in the recursive sampling scheme; It is just a DGP model with the doubly stochastic inference framework slightly modified to only consider parents of a vertex in the previous layer during updates/sampling. The main Theorem is somewhat trivial as it is against a standard DGP that does not use any of the graph information hence it has full dependency (and hence larger resulting sampling variance) whereas by slightly restricting the dependency structure (via a 1st order Markov assumption on parents as in DGPG) you can reduce that variance. It is somewhat obvious that this is the case. All comparisons on real data are against Graph ANN approaches and [20] on the small dataset; It would help to include probabilistic deep learning models such as a Deep Gaussian Markov Random Fields (ICML 2020, arxived since February) or a Bayesian ANN approach or a Convolutional Gaussian Process or [18] or or [19] Graph kernels, that would better exploit the graph structure.

Correctness: As far as I examined its a correct adaptation of the DGP model and doubly stochastic inference to work with Graphs but it is somewhat trivial. Empirical results don't compare against other DGP/Deep GMRF/Bayesian graph ANN/Convolutional GP methods.

Clarity: Yes

Relation to Prior Work: Somewhat yes, although Deep GRMFs [ref above], standard GMRFs [Rue & Held] and Convolutional GPs, (and [18, 19]) are very relevant to such graph structures, as is the general area of Graph Kernels [Journal of Machine Learning Research 11 (2010) 1201-1242].

Reproducibility: Yes

Additional Feedback: Summarising from above, I think the paper is technically correct but lacks in novelty and offers a minor adjustment of a standard DGP and corresponding doubly stochastic inference method to encode the graph structure. The novelty and contribution offered is contained in upper half of page 5 and its a straightforward modification to exploit parts of the graph structure within a DGP and resulting ELBO. Unfortunately I don't have any good ideas on how you would improve the novelty further, but perhaps looking at related work and the connections to GMRFs and Graph kernels would help to at least explore connections/analogies and to improve the experimental comparisson. Also, I don't think the ICASSP paper [20] (which again and similarly to this paper only compares with a standard GP) is the state of the art for this problem in GP land, but looking at [18] and the aforementioned literature on Graph Kernels, and (Deep) GMRFs is. In the synthetic experiment I have specific questions for the authors: You are training a DGPG with 1 layer, is the neighbourhood used in the DGPG defined to be the same 7 parents that you have used to construct the output data of each node? If so, isn't it obvious that a DGP which is closer to the data generating process by construction (encodes that neighbourhood dependency), would perform better and have less variance from another DGP which tries to model a larger dependency structure (that does not exist by construction)? Isn't the conclusion hence trivial? Finally, the main focus of comparison on the probabilistic side is between a general vanilla DGP that is completely ignorant of the graph structure, and the proposed Graph DGP version. If I have a fully connected graph, is the GDGP equivalent to a DGP and hence just a special case of it? In rebuttal I invite the authors to convince me to raise my score by focusing on two key points mentioned above in various sections and questions: 1) The novelty of the approach and the strength of the contribution in relation to the underlying DGP model and inference scheme 2) The experimental comparison is against state of the art GP/probabilistic approaches for Graphs as I am not convinced that running [20] on the small dataset is convincing. Minor: - various spelling/grammar mistakes like L35 natural->nature, L147 edged-edges, etc; check again - ARD - I thought initials are Automatic Relevance Determination, not "Discovery" - Line 89: "In [11]...(DGP)" Thats wrong, the correct reference is [7] where the DGP was first introduced and in the accompanying PhD thesis by Damianou. - Line 66-67: "..assuming both the prior and the likelihood .. are Gaussian" Thats also not true, a GP is a prior over functions and IF the likelihood is Gaussian you have GPR which is closed form. GPs don't assume the likelihood is Gaussian necessarily. - Why you need the word "Stochastic" in the method/title? a GP is a stochastic process eitherway so it does not add any new info? Are you trying to highlight that inference in the model follows the doubly stochastic framework? Its not needed. - All the metrics in experiments are focusing on accuracy (MAE/MAPE/RMSE) and none capture uncertainty quantification which would be the main benefit of a Bayesian approach like a (D)GP. ## Post-rebuttal I have read the author's response and appreciate the responses and additional study on corrupting graph connectivity. I do appreciate that modifying a GMRF to compare against in their examples might not be as trivial as I thought. However, I still think this work is a special case of a DGP - as the authors agree that if graph is fully connected you get back the DGP model and associated doubly stochastic inference - with limited novelty on model or inference beyond just restricting connectivity/dependency in the model. More importantly the paper would benefit with a comparison against a proper probabilistic kernel-based graph approach. You are proposing a probabilistic kernel-method for graphs without comparing against the state of the art in this specific area e.g. DGP & GP with proper graph kernels (not just linear/rbf/matern). Given the limited technical/model/inference novelty, the empirical comparisons should be at least complete and compelling for NeurIPS. Graph kernels where introduced a decade or more ago. I will not change my score of 5 and if the paper gets accepted I would strongly encourage inclusion of comparison against GPs & DGPs with graph kernels at least which should be trivial to do.


Review 4

Summary and Contributions: Stochastic Deep Gaussian Processes over Graphs The manuscript proposes a variant of the stochastic deep Gaussian processes (DGP) which uses the graph structure (encoded in input and output signals) to define the conditional independence structure of the DGP priors (in eq.3, F^{l,k} is independent from the unconnected nodes in layer l-1 given F^{l-1,pa(k)}). The proposed model can be learned by the recursive sampling method which is modified from that of the original doubly stochastic DGP paper [11].

Strengths: Significance and relevance: Using the graph structure in DGP is a quite interesting idea, and the proposed method outperforms the previous GP-based work (which also is based on the graph structure) and shows comparative performances to a state-of-the-art-method (DCRNN) for the nontrivial prediction problem. I think that the manuscript shows a good application of DGP to real-world problems.

Weaknesses: Novelty & contributions 1. My main concern is that the novelty and contributions of the proposed method do not seem to be significant. It seems to me that the proposed method is a direct application of the stochastic DGP [11] to a certain problem domain where input and output signals have the graph structure. All the inference steps appear to be straightforward from the previous work [11].

Correctness: I think the method is technically correct.

Clarity: I think that certain points are unclear and need to be modified. - In line 148, I think that the term ‘parents’ can be a little bit misleading. This term refers to just connected nodes (neighbors). The term ‘parents’ is more commonly used for directed graphs (e.g., trees or directly acyclic graphs) than for undirected graphs. I think that “neighbors” of a given node would be enough (and more appropriate). I think that the proposed method mainly utilizes the locality information from the graph to make prediction (in eq. 3, functions values at the node k in the layer l are dependent on those at the parental nodes of the node k in the l-1 layer). Thus, it does not make a sense to me that the parent node set pa(v_k) does not include v_k itself (based on Figure 1, parents nodes pa(v2), v_1 and v_3, do not include v_2). For example, a single layer DGP (L=1), the first term of the GP prior in eq. 3 becomes p(F^{1,k}| U^{1,k}; X^{pa(k)}, Z^{pa(k)}), and it is more reasonable that the GP prior defined on the node k should involve the features associated with the node k. - In the experimental result section, it is not clear about what the datasets look like. A table containing the basic statistics (number of samples, dimension, the average number of neighbors of a node in the graph, and so on) of the datasets should have been included. In addition, it is not clear about the sentence in the lines 232-233. It seems that all the small (and large) datasests are not graph datasets because the authors mentioned in the line 234 (and 253) that they built an adjacent matrix for each dataset. Is it correct? I think that the authors should include further explanations about the datasets and how to build graphs from them (e.g., how to define the distance between two nodes?).

Relation to Prior Work: I think that the authors should more clearly explain the previous work based GP [20] and how the proposed method differs from it (because this work is one of main baseline methods)

Reproducibility: Yes

Additional Feedback: Typo: in the line 208: DGPG should be DGP?

[Author Response · NeurIPS 2020]

Table 1: $r$ decreases as the graph is corrupted.

| Run | $p=0$ | $p=2$ | $p=4$ | $p=6$ |
|---|---|---|---|---|
| 1 | 4.8E18 | 2.9E17 | 2.8E15 | 6.1E13 |
| 2 | 4.8E18 | 2.8E16 | 3.3E15 | 4.2E13 |
| 3 | 4.8E18 | 1.1E16 | 6.8E14 | 4.5E13 |

Table 2: Additional results on small datasets (RMSE).

| Dataset | DGPG-L | SVR | RT | MLP |
|---|---|---|---|---|
| Weather | 1.92 | 1.88 | 2.35 | 2.06 |
| fMRI | 0.020 | 0.020 | 0.021 | 0.024 |
| ETEX | 0.41 | 0.31 | 0.31 | 0.31 |

**Usefulness of the Theorem:** We perform a new experiment to show the usefulness of our theorem. It has been showed that $\alpha \gg \beta$ implies the algorithm benefits from graph information and thus converges faster. Now we use $r = \alpha/\beta$ to measure the richness of information of graphs: larger $r$ suggests more informative graph structures. We corrupt the graph by randomly select $p = 0, 2, 4, 6$ edges and change them into erroneous connections. As $p$ increases $r$ becomes smaller (corrupted graph is less informative), which follows our analysis and expectation. Our theorem and the statistics it derives have potential applications, e.g. determining the network structures. We show 3 runs due to space limit.

We summarize the significance of the theorem as follows: 1) It presents a rigorous proof that considering graph structure can reduce sampling variances and thus encourage convergence; 2) Our additional experiment shows that the statistics developed in our proof have potential applications, e.g. determining more probable network structures; 3) Though the same goal can be achieved by observing ELBO in this example, they are derived from totally different mathematical principles. Their evaluation processes are also different: while evaluating ELBO requires stochastic sampling with the "re-parameterization trick", our statistics can be computed by algebraic manipulation and is more favorable; 4) One may further wander if it is possible to optimize $\alpha/\beta$ together with ELBO, which we leave for future investigation.

**Additional Experiments on Small Datasets:** We only present RMSE due to space limit. DGPG-L uses linear kernel (suggested by Reviewer #2); We further compare with standard regression algorithms (support vector regression, regression tree and MLP) where the output is a function of its neighbors' input (high-order graph information is lost).

**Key Novelty and Contributions:** 1) Though our method shares similarities with [11], optimizing ELBO with stochastic recursive sampling is a novel attempt and could potentially be extended to other structural domains (e.g. image). 2) Our theoretical analysis on the sampling variance is novel, introducing new insights and understandings. 3) Graph analysis community can better benefit from our work rather than from [11]: DGPG can model uncertainties and determine the importance of the connections with ARD kernels, which are important for graph analysis. 4) Our implementation is nontrivial: it transforms the graph into parallelizable data structures, and takes advantages of GPU acceleration. Both graph analysis and GP communities can benefit from our code, which will be available in our public repository.

**Response to Reviewer #1:** 1) We considered the evaluation of posterior variances in the experiment of Large Dataset (Section 4.3). Table 4 shows that our method can model uncertainty with satisfactory accuracy. 2) Units in Table 2 & 3 are different for each dataset. For instance the unit in Weather domain is Celsius. 3) An advantage of DGPG is that it does not require validation set. Train/valid/test splitting in the traffic flow prediction dataset is about 7/1/2, the same as [45] for comparison. DGPG can use the validation set for training as we discussed in Line 256-258. 4) Inducing points are optimized. 5) Number of iterations in the Weather/fMRI/ETEX/traffic datasets are 2000/5000/5000/10,000.

**Response to Reviewer #2:** 1) Usefulness of Theorem 1: Please see the discussions at the beginning. 2) DGPG with linear kernel: Please see the additional results in Table 2. 3) The number of inducing points is mostly chosen empirically. We would recommend to use $M = \sqrt{0.1N}$ where N is # of training instances, which is approximately the parameters we used. We find this value does not have too much effect as long as it is in a reasonable range. Initial locations of the inducing points are obtained by running k-means on the training data. 4) We thank the reviewer for pointing out the relevant work of [Bui] and [Burt et al.]. We will use the results from [Burt et al.] to enrich our discussion on complexity.

**Response to Reviewer #3:** 1) We hope that the reviewer's concern about novelty, contributions and comparison with stronger baselines can be addressed by our discussions and new experiments presented at the beginning. 2) [18], [19] and recent works of Bayesian graph ANN do not share the same goal of our paper, e.g. while DGPG learns from multiple signals some of their work consider 'node prediction' where only one signal is available and some nodes' values are missing. There is no obvious way to apply them to the datasets we consider. 3) GMRFs are a powerful model for interpolating signals over graphs. However our experiments require extrapolation across vertices (e.g. predict 90 nodes from 10). Preliminary results show that they are not suitable for comparison. We will review them in our discussion of related work. 4) In the new experiment the graph is no longer "ground true": we corrupt the graph with erroneous connections and analyze the results. 5) When the graph is fully connected, DGPG reduces to DGP. DGPG is a more generalized model, and considering the graph structure also brings novel analysis and insights.

**Response to Reviewer #4**: 1) Novelty & contributions: Please see the discussions at the beginning. 2) DGPG can be applied to directed graph so 'parent' is a more suitable term, e.g. in the traffic dataset the adjacent matrix is asymmetric. 3) In all current experiments $pa(v_k)$ include $v_k$. But it is still possible that the output signal does not depend on the input signal of the same node, so we leave it a task-specific choice whether the self-connection would be modeled. 4) We will describe the basic statistics that you mentioned in the table. In all the experiments each vertex has location information, and the graph is constructed according to Euclidean distances. We will explain this explicitly in our paper.

We thank all the reviewers for their insightful comments, which will be properly addressed in our later version.

[Meta-Review · NeurIPS 2020]

This paper shows good empirical results on graph learning. While some reviewers would have preferred to see a larger contribution on the technical part (see e.g. useful suggestions by R3), it was deemed that the demonstration of a DGP working convincingly on graphs was significant enough without complicated technical derivations such as those often seen in DGP papers (which focus on new inference methods rather than new application areas). Importantly, the motivation and practical merit of combining DGPs with graphs has been made clear: it allows to perform deep learning on graphs (a recent, successful trend in the field) with added and well-analyzed qualitative advantages, namely obtained variance and ARD relative importance.